# Understanding Mental Health in Developmental Dyslexia: A Scoping Review

**DOI:** 10.3390/ijerph20021653

**Published:** 2023-01-16

**Authors:** Adrienne Wilmot, Penelope Hasking, Suze Leitão, Elizabeth Hill, Mark Boyes

**Affiliations:** 1Faculty of Health Sciences, Curtin School of Population Health, Curtin University, Perth, WA 6845, Australia; 2Faculty of Health Sciences, Curtin enAble Institute, Curtin University, Perth, WA 6845, Australia; 3Faculty of Health Sciences, Curtin School of Allied Health, Curtin University, Perth, WA 6845, Australia

**Keywords:** dyslexia, reading difficulties, mental health, internalising, externalising

## Abstract

Children with dyslexia are at elevated risk of internalising and externalising mental health concerns. Our aim was to scope the extent and nature of the literature investigating factors which may influence this association. We systematically searched the peer-reviewed and grey literature with no restrictions on the date. We included both qualitative and quantitative studies. Inclusion criteria included: (1) a focus on childhood (≤18 years) reading/learning difficulties; (2) internalising and/or externalising symptoms; and (3) a potentially modifiable third factor (e.g., self-esteem). Ninety-eight studies met the inclusion criteria. We organised the studies according to individual, family, and community-level third factors. Whilst a range of third factors were identified, relatively few researchers tested associations between the third factor and mental health in the context of dyslexia. Furthermore, there was a focus on primary rather than secondary school experience and a reliance, in many cases, on teacher/parent perspectives on children’s mental health. Future researchers are encouraged to explore links between socio-emotional skills, coping strategies, school connectedness, and mental health in the context of dyslexia. Research of this nature is important to assist with the identification of children who are more (or less) at risk of mental health concerns and to inform tailored mental health programs for children with dyslexia.

## 1. Introduction

Dyslexia is characterised by difficulties with accurate and fluent word reading and poor spelling and decoding abilities that do not progress as expected with the provision of well-intentioned and targeted intervention [1]. Importantly, dyslexia is not related to more generalised cognitive difficulties or sensory deficits, rather, the difficulties are thought to stem from neuro-cognitive differences in the way speech sounds are processed [2,3]. Consistent with this account, oral language difficulties in early childhood are a frequent precursor to dyslexia [4] and difficulties in both the oral and written (reading, spelling, writing) domains of language often co-occur across the lifespan [5,6,7]. Furthermore, dyslexia is frequently associated with other learning and attentional difficulties that are believed to share genetic risk factors and/or underlying cognitive differences [8,9].

The *Diagnostic and Statistical Manual of Mental Disorders-Fifth Edition* (DSM-5); Ref. [10] has a category for dyslexia within the Specific Learning Disorders (SLD); a category which also includes dyspraxia (difficulties with writing) and dyscalculia (difficulties with mathematics). It is difficult to estimate prevalence rates for dyslexia as many children remain unidentified. Nevertheless, it is generally believed that 5–10% of children experience severe and persistent word reading difficulties consistent with dyslexia; a figure which equates to approximately 2–4 children in an average-sized classroom [3,11].

### 1.1. Academic and Psychosocial Correlates to Dyslexia

Dyslexia can negatively affect children’s motivation to read [12], vocabulary acquisition and reading comprehension. These skills are important for higher level learning and more complex academic content as the school years progress. Qualitative accounts suggest that children with dyslexia are often aware of the extra effort they need to put into their studies to achieve at comparable levels to their peers [13,14]. Similarly, longitudinal evidence suggests that many, but not all, experience relatively poor educational and occupational outcomes [15,16]. Furthermore, those who live and work with children with dyslexia frequently report concern for children’s mental health [11,14,17]. Indeed, there is considerable evidence to suggest that dyslexia is associated with a range of psychosocial difficulties in childhood including: reduced academic self-concept [18], poor reading self-efficacy [19], and elevated levels of internalising (e.g., anxiety) and externalising (e.g., aggression) symptoms indicative of poor mental health [20,21,22]. Taken together, these findings suggest that the mental health concerns of children with dyslexia can extend beyond the classroom into their everyday lives and may persist into adulthood. Understanding *why* children with reading difficulties, such as dyslexia, are at elevated risk of mental health concerns is now a stated research imperative [22,23]. 

### 1.2. The Current Review

The aim of the current review was to scope the extent and nature of the literature investigating factors which may influence the association between childhood dyslexia and internalising and externalising mental health concerns. In so doing, we aimed to identify gaps in the literature and use the findings of the review to inform decisions about directions for future research. Such research is needed to: (1) identify children who may be more (or less) at risk of mental health concerns, and (2) inform tailored mental health programs. Given this aim, a scoping rather than a systematic review was indicated. (For the purpose of this review, mental health concerns are defined according to an internalising/externalising dichotomy [24,25]. which is widely used in the child development literature and clinical settings. Internalising refers to expressions of emotional distress which are inwardly focused (e.g., withdrawal) whilst externalising refers to expressions of emotional distress which are outwardly focused (e.g., aggression).

A preliminary search for existing reviews and meta-analyses on this topic was conducted in April 2021 using the following databases: Prospero, Cochrane Database of Systematic Reviews, JBI Evidence Synthesis, Campbell Collaboration, and the Open Science Framework. We were unable to find any planned or existing reviews with the same focus as ours. Previous reviews have focused on investigating the strength and direction of the relationship between learning/reading difficulties and internalising symptoms [21,22,26] or learning difficulties and both internalising/externalising symptoms [27]. Another review by Haft and colleagues [28] offered an excellent preliminary discussion of protective factors for socio-emotional resilience in the context of developmental dyslexia but did not comprehensively search the literature of the time. Our review therefore offers an extension to the existing field.

We came to an a priori decision to present our findings using the three-tiered framework of (1) individual, (2) family, and (3) community-level factors developed by Haft and colleagues [28] and applied previously in research on dyslexia and child mental health [29]. This framework acknowledges that children’s mental health is influenced by a range of factors that relate to the child themselves (the individual), their family, and the communities in which they participate.

## 2. Materials and Methods

Our review was informed by the Joanna Briggs Institute (JBI) guidelines [30] and the PRISMA extension for scoping review checklist [31] which build on Arksey and O’Malley’s [32] foundational work on scoping review methodology. A protocol for the review was registered on the Open Science Framework on 04.05.2021 and is available to view at 10.17605/OSF.IO/GZ98X. Changes to the protocol that were applied during the review process are described in Appendix B.

### 2.1. Inclusion/Exclusion Criteria

*Types of studies.* Peer-reviewed primary research studies as well as theses/dissertations were considered for inclusion. When the same research was reported in a dissertation and journal article then the dissertation was excluded. No limits by date were set. Books, book chapters, study protocols, theoretical/opinion pieces, and previous reviews were excluded.

*Population.* Sources that included children (≤18 years) or adult perspectives (e.g., teachers, parents, or adult retrospective studies) on children’s mental health were considered so long as the child participants spoke an alphabetic language and were not being tested in a language other than their primary language.

*Concept 1: Reading difficulties.* Sources were included if they investigated internalising and/or externalising symptoms among children with recognised learning difficulties and/or word reading difficulties consistent with dyslexia. In keeping with DSM-5 diagnostic criteria, when the learning difficulties could be better explained by medical (e.g., hearing loss), neurodevelopmental (e.g., autism, intellectual disability), or socio-cultural factors (e.g., child poverty, second language learners, lack of educational opportunity) sources were excluded [10].

*Concept 2: Mental health.* Mental health concerns were defined as consisting of both/either internalising or externalising symptoms. Sources which used global measures of psychosocial functioning with sub-scales to measure internalising/externalising symptoms (e.g., Strengths and Difficulties Questionnaire (SDQ); Goodman, 1997 [33]), as well as those which measure a particular aspect of internalising/externalising symptoms (e.g., anxiety) were included. For qualitative research, self/parent/other description of emotional difficulties or mental health concerns were sufficient for inclusion.

*Concept 3: Third factor.* Sources were only included if they investigated a modifiable “third factor” in association with mental health concerns among children with learning/reading difficulties. By modifiable we are referring to a skill/attitude/behaviour (e.g., emotion regulation) which could become an intervention target for mental health programs designed for school-aged children with reading difficulties and/or their families and communities. For this reason, sources which solely investigated factors which are intractable (e.g., a child’s age); factors which cannot be modified once a child has reached primary school age (e.g., exposure to environmental toxins in-utero); or cognitive differences that may be difficult to modify (e.g., working memory, attention) were not included.

*Context.* Sources were not excluded based on geographical factors, the educational context of the children (e.g., mainstream, or specialised school), or whether children were receiving educational support or socio-emotional or reading intervention.

### 2.2. Search Strategy

An initial set of search terms were developed by the research team based on their current understanding of the literature. A 3-step search strategy was then followed as recommended by the JBI [30] (p. 418). At the first stage, the first author conducted a preliminary search of two databases (Scopus and ProQuest) on 10.06.2021. Based on the subject headings and keywords from relevant sources found in this preliminary search, an additional 4 keywords were added to the search terms (Appendix C).

The first author then conducted the second stage of the search on 11.07.2021 with an update on 17.05.2022. This included a search by subject heading and keyword of the OVID platform databases: Embase, PsychInfo, and OVID Medline, and a search by keyword on Scopus, ProQuest and CINHAL. Limits were set for English language and to peer-reviewed journals (where possible). The first author then conducted a search for dissertations/theses using the same keywords via the ProQuest Dissertation and Thesis database. See Appendix A for an example of a complete search of one database, PsychInfo. For the third and final stage of the selection process, the first author reviewed the reference lists of key sources and relevant reviews. An additional 8 sources were located at this stage.

### 2.3. Source Selection

A total of 10,810 sources (10,155 articles and 655 theses) were downloaded into Endnote [34] and duplicates removed by the Endnote de-duplication process and by hand. This resulted in a total of 7389 sources (6840 articles and 549 theses) being considered by title or title/abstract from the database search (a further 8 sources were added at a later stage from the third stage of the search). At this stage, a source selection tool (see Appendix A) was developed and piloted by each member of the review team on a random selection of 25 sources. No one reviewer achieved less than 88% agreement with the first author and 4 members of the review team of 6 achieved 100% agreement with the first author. After this pilot, no changes to the subject headings or keywords of the search were considered necessary. However, the source selection tool was adjusted through team discussion to add further explanation and examples to aid the selection process. Throughout the source selection process, we further refined the inclusion/exclusion criteria as outlined in Appendix B.

Following this, a random sample of approximately 20% of sources was independently reviewed by two members of the review team (the first author and one other) by title and/or title/abstract. An inter-rater agreement of 88.6% was achieved. This was considered a satisfactory level of adherence to the selection criteria by the first author who then proceeded to select sources for consideration from the remaining 80% of total sources. Disagreement between reviewers was resolved by consensus by two members of the review team.

A similar process was followed for the next stage of the selection process; selection for inclusion/exclusion based on full text review. Approximately 20% of sources were independently screened by two members of the review team (the first author and one other) (inter-rater agreement of 82%) and the remainder was reviewed by the first author alone. Disagreements between authors, and/or any uncertainty by the first author, were resolved by group consensus. A record of the number of sources included/excluded at each stage of the source selection process and the reasons for exclusion are included below in Figure 1.

### 2.4. Data Extraction

A data extraction form for this review was adapted from models presented by the JBI manual and other authors [30,32] and developed with input from 3 members of the research team. The first author extracted the relevant data from each source and the fourth author (EH) checked the data extraction of 20% of sources. Based on EH’s review, one source was excluded and the participant details of another were corrected.

## 3. Results

A total of 98 studies (12 theses and 86 peer-reviewed journal articles) met the inclusion criteria. Geographically, the scope of the studies was widespread. A total of 20 countries were represented; the largest numbers of studies originated from the USA (32), Italy (11), United Kingdom (10), and Israel (7). One study was a cross-cultural analysis of children’s mental health and coping strategies and included participants from Germany and Indonesia [35]. In terms of design, nine studies were qualitative, 10 incorporated a mixed methods approach, and the remainder were quantitative by design, incorporating either a longitudinal (21 studies) or cross-sectional (58 studies) approach. One of the longitudinal studies was a secondary analysis of four longitudinal studies from English-speaking countries [36].

### 3.1. Sampling Characteristics

Approximately one quarter of the included studies (27 studies; 27.5%) were not dyslexia specific, that is the researchers conducted their investigations using samples of children/adolescents with learning difficulties without specifying sub-type (reading, writing, mathematics difficulties) or specified that the children all had a diagnosis of SLD mixed-type (difficulties in at least two domains of reading, writing, and/or mathematics), see for example [37]. Researchers of the remaining 71 studies focused on developmental dyslexia or word reading difficulties consistent with a dyslexia diagnosis. A wide range of methods were used to determine group status, ranging from self-identification of reading difficulties, see for example [38], to extensive diagnostic and validation measures, see for example [39]. Furthermore, the severity of dyslexia (when reported) ranged from mild [40] to severe [41].

Sample sizes ranged from one (a case study) [42] to a study incorporating findings from four databases totaling 23,799 children [36]. In terms of the age and the stage of development of participants, the researchers of 46 studies focused exclusively on primary school-aged children (≤12 years; 46 studies); 17 on adolescents (≥12 years); and 27 included samples of young people whose ages ranged across these developmental periods (childhood–adolescence). Of the remainder, there were four longitudinal studies in which children were tracked from their primary to secondary school years [43,44,45,46]; three studies in which adults were interviewed about their childhood experiences with dyslexia, see for example [47]; and one study in which educators of children with dyslexia aged 5–18 years were interviewed [11]. In many of the longitudinal studies the researchers stopped assessing children in the primary school years, see for example [36,48,49,50,51,52,53,54,55,56,57,58].

### 3.2. Study Aims and Mental Health

There was great variety with regard to the aims of the included studies. For example, some researchers primarily aimed to investigate the relative contribution of SLD or attentional difficulties (e.g., co-morbid ADHD) in explaining internalising/externalising symptoms, see for example [59]. Many researchers had the primary aim of investigating the direction of effects between learning/reading difficulties and a range of psychosocial (e.g., self-esteem, social skills) variables, see for example [36,50,54,55,56,60,61]. In other studies, mental health was not included as an outcome variable. Rather, the relative influence of learning difficulties and/or behaviour/mental health on children’s social skills [62] or academic outcomes was investigated (e.g., grades, reading comprehension, secondary school completion), see for example [43,57,63,64]. However, in our estimation, the aim of 39 (40%) of the included studies aligned to that of this review; that is, the researchers aimed to better understand mental health in the context of childhood learning/reading difficulties. Not all researchers found dyslexia-related group differences in mental health concerns, see for example [65]. However, when differences were found, the balance of evidence suggests that reading difficulties precede internalising symptoms [36,46,48,50,52,54] whereas the direction of effects between dyslexia and externalising symptoms was less certain. Specifically, there is evidence that externalising symptoms are associated with attentional difficulties which may be present before school entrance but can worsen in response to school demands [46,66,67].

In the qualitative studies (and one mixed method study), mental health was investigated among children with reading difficulties broadly. Among the studies incorporating a quantitative approach, the researchers of 25 studies focused on internalising symptoms only and those of one study focused on externalising symptoms only [59]. Most researchers in this review (62; 63.3%) investigated both internalising and externalising symptoms. Many conducted their investigations of internalising/externalising symptoms with well-validated measures of global psychosocial functioning such as the Strengths and Difficulties Questionnaire (SDQ) (10 studies) and the Child Behaviour Checklist (CBCL); Achenbach (various versions, for example, [24]) (27 studies). Of the total sample of the included studies (qualitative and quantitative), the researchers of 39 studies combined perspectives (e.g., child and/or teacher and/or parent and/or both parents) on children’s mental health. However, in nearly half of the included studies (45; 45.9%, excluding the adult retrospective studies where adults reflected on their own childhood experiences) children’s perspectives on their mental health were not considered. Furthermore, a wide variety of child-reported measures of anxiety (11 measures) and depression (10 measures) were included in this review.

### 3.3. Third Factors

The research team formed 26 third factors categories (see Table 1 below) on the basis of common tools or measures and definitions provided in the publication. Each category was allocated to individual, family, and community-level factors. Many studies investigated factors on more than one of these levels. Where constructs are included in parathesis, they were deemed to be related to the primary third factor. Studies have been placed in a column depending on whether they primarily dealt with the third factor using qualitative, quantitative or both approaches. When a quantitative approach was taken, we have provided an indication of whether a statistical association between the third factor and the mental health concern in the context of learning difficulties was found, not found, or was not tested. Appendix A presents the summary of findings from all 98 studies included in this review including information regarding the direction of association between the variables.

## 4. Discussion

The aim of the current review was to scope the extent and the nature of the literature investigating factors which may influence the association between childhood dyslexia and internalising and externalising mental health concerns. In so doing, we aimed to identify gaps in the literature and use the findings of the review to inform decisions about directions for future research. Such research is needed to (1) identify children who may be more (or less) at risk of mental health concerns and (2) inform tailored mental health programs. To this end, our review identified a total of 98 sources (86 peer-reviewed journal articles and 12 theses) for inclusion dating from 1968. Twenty different countries were represented highlighting a global concern for the mental health of children who struggle to read. Our review includes: 70 studies in which individual-level factors (e.g., self-esteem) were explored; 39 in which family-level factors (e.g., the parent-child relationship) were explored; and 54 in which community-level factors (e.g., the teacher-child relationship) were explored. Most researchers investigated both internalising and externalising symptoms but those of 25 studies focused exclusively on internalising symptoms. One explanation is that internalising symptoms, specifically anxiety, have been highlighted as a particular mental health concern among children with dyslexia, see for example [21,22].

### 4.1. Social Experiences

The social skills (20 studies) and the social problems (29 studies) of children with word reading/learning difficulties were amongst the most studied “third factors” in this review. However, social difficulties were most often studied as correlates of learning difficulties, see for example [48,50,110], or poor academic performance [62] or attentional difficulties [59,129] that may co-occur, rather than as possible risk/protective factors for mental health. Furthermore, there was a focus on investigating children’s challenges (e.g., peer difficulties) from the parent and/or the teacher perspectives rather than their strengths. Children’s subjective feelings of loneliness were also under-studied, a finding consistent with Kwan and colleagues’ (2020) recent review [130]. Nevertheless, our review highlighted consistent links between bullying involvement and mental health concerns [29,58,124,125] and the protective function of friendship, see for example [77,117], in the context of childhood dyslexia. Differences between child, parent, and teacher reports of social difficulties were reported by some researchers, see for example [41,61], highlighting the importance of examining the context of social difficulties (e.g., school or home) in addition to gauging children’s own perspectives on their social strengths and challenges in future research.

### 4.2. Self-Esteem

Other individual-level third factors which have been widely studied in this field are self-esteem (and related constructs, e.g., self-concept/self-efficacy) (30 studies) and stress, coping and resilience (18 studies). A secondary analysis of four different longitudinal studies by McArthur and colleagues (2022) found a link between early reading difficulties and later anxiety, depression, and poor reading self-concept (beliefs about oneself as a reader) [36], suggesting that poor reading self-concept may be a risk factor for anxiety and depression. Consistent with this, Terras and colleagues (2009) found an association between low scholastic self-esteem (beliefs about oneself as a learner) and parent/child reported internalising symptoms [73] and Giovagnoli and colleagues (2020) found that adolescents with dyslexia who reported low levels of self-efficacy (belief in one’s ability) with regard to school tasks experienced more somatic (headaches, stomach aches) symptoms [39]. However, there were some mixed results across the field regarding both the domain of mental health (anxiety, depression, externalising) and the domain of self-esteem/self-efficacy (e.g., scholastic self-esteem, social self-esteem) involved. Furthermore, most studies which examined the associations between self-esteem and internalising/externalising symptoms were cross-sectional by design meaning that the direction of effects could not be determined. Disentangling the strength and direction of these associations, and factors that support children’s self-esteem in the context of dyslexia, would allow for the provision of timely and targeted support for children at risk of mental health concerns.

### 4.3. Coping and Resilience

Although over a decade old, Singer’s (2005, 2007) foundational work suggests that the strategies that children with dyslexia use to cope with school-related difficulties (such as teasing and poor grades) are instrumental in supporting their self-esteem (or not) [68,69]. For example, the children she interviewed used self-talk to either support or hinder their self-esteem. Consistent with this, Hossein and colleagues (2022) found that children with reading disorder who had more “grit” (i.e., perseverance) and “resilience” were less likely to experience anxiety (reported by teachers) and depression (reported by parents) [91]. Similarly, Giovagnoli and colleagues (2020) found that internalising symptoms among adolescents with dyslexia were associated with a tendency to react to school-related problems with hypervigilance, defined as an “excessive sense of alertness” [39] (p. 461). However, two other groups of researchers in this review did not find any group differences in coping related to learning/reading difficulties although higher levels of internalising symptoms were observed [35,94]. As such, we believe that an investigation of coping strategies, especially with regard to the school context, in relation to children’s mental health warrants further exploration.

### 4.4. Emotion Regulation and Academic Factors

Emotion regulation (the ability to understand and manage one’s emotions) is one aspect of coping which appears to be relatively under-studied in this field. Several studies in this review included an examination of factors related to emotion regulation, such as emotional intelligence, see for example [99,106]; focused on one aspect of emotion regulation, namely rumination [97]; or measured emotion regulation with a single item on a parent survey [29]. As a result, we believe that emotion regulation in the context of dyslexia has not been comprehensively explored. This represents a gap in the literature given that emotion regulation is strongly associated with mental health across the lifespan [131,132,133] and may be hindered in children who experience language/literacy difficulties in early childhood and beyond [134,135,136]. Certainly, there is evidence from experimental research that children with dyslexia, relative to controls, have more difficulty with recognising emotions in others (from facial and vocal cues) [137,138]. This may suggest difficulty with understanding their own emotions, a basic building block to emotion regulation [139]. Furthermore, many children with reading difficulties report experiencing heightened negative emotions (e.g., frustration, anger, sadness) in the school context [14,55] suggesting that effective emotion regulation may be a particularly salient intervention target for their mental health. Longitudinal studies using well-validated (child and parent/teacher versions) measures of specific socio-emotional competencies (e.g., emotion regulation, spoken language skills) are needed to test these associations over time and to build upon the existing work described in this review. Relatedly, children’s attitude to their learning and whether they succeed in their studies despite their learning challenges has been linked to their mental health, see for example [57,65,108,109], and warrants replication.

### 4.5. Family Factors

In terms of family factors, the association between children’s mental health and parental psychological variables (e.g., parental stress, anxiety, self-esteem, coping, emotion regulation); the quality of the child-parent relationship; family functioning/support; and parenting values/practices is well researched. A strength of this field is the widespread use of well-validated measures such as the Parenting Stress Index (PSI) and its short form [140] enabling a comparison of the findings across the field. Importantly, the PSI-SF has been validated for use with parents of children with a wide variety of mental, emotional, and behavioural difficulties [141]. However, much of the research in this section of the review has employed samples of children with a wide range of learning and attentional difficulties without specifying the sub-type, see for example [98]. This is a limitation given that parents of children with dyslexia may have unique strengths and challenges, see for example [83,113]. Furthermore, the perspectives of fathers and siblings were often absent and there are mixed results which require further exploration. Nevertheless, future researchers are encouraged to investigate the value of whole-family support within mental health promotion programs for children with dyslexia and other learning difficulties.

### 4.6. School and Community Factors

Certainly, the importance of having parents who understand dyslexia and provide both emotional and academic support was highlighted by many studies in our review, see for example [14,73,117]. However, when interviewed, people with lived experience of dyslexia (especially parents/teachers and older adolescents/young adults) also raised school/community-level concerns such as: teachers’ misunderstanding of dyslexia; a lack of accommodation and support for reading-related challenges; and experiences of stigma, shame, and discrimination in learning environments, see for example [11,14,42,47,121]. Findings such as these suggest that children with dyslexia and other learning difficulties may be vulnerable to low levels of “school connectedness”, a concept that describes perceptions of being understood, supported, and treated fairly at school (by peers and by teachers alike) [142,143]. Consistent with this, two sets of researchers in this review found evidence to suggest that school connectedness may be a particularly salient protective factor for the socio-emotional wellbeing of children with learning difficulties [107,126]. Similarly, Chiappedi and Baschenis [122] found that children with SLD who believed that their teacher understood and supported their learning disability reported significantly lower levels of anxiety. Future researchers are encouraged to replicate the aims of these studies using well-validated measures of school connectedness and dyslexia-specific samples. There is evidence from our review that children with dyslexia experience a different trajectory of socio-emotional difficulties during the school years than those with other special educational needs, see for example [50], and therefore need to be differentiated in future research.

### 4.7. Methodological Considerations

A strength of the field is the widespread use of well-validated measures of children’s psychosocial functioning such as the Strengths and Difficulties Questionnaire (SDQ)and the Child Behaviour Checklist (CBCL).This aids cross-cultural comparisons and the generalisability of each study’s findings. However, many studies included in this review examined group differences (e.g., children with and without reading difficulties) with regard to the third factor rather than testing associations between the third factor and mental health concerns. This limits our understanding of risk and protective factors for mental health. Furthermore, in many cases, researchers relied on parent/teacher, rather than child, perspectives on internalising symptoms, see for example [49,126]. Internalising symptoms may be difficult for an outsider to accurately assess; therefore, future research which gauges children’s own perspectives is encouraged. Furthermore, to advance understanding of risk/protective factors for specific (e.g., depression) mental health concerns then the use of well-validated measures which align to DSM criteria (such as the Revised Children’s Anxiety and Depression Scale (RCADS; Chorpita et al., 2000) [144] and include parent/child versions, rather than (or in addition to) broad-band measures of psychosocial functioning such as the SDQ, are needed.

Additionally, our review highlighted a bias in the field towards investigating the psychosocial wellbeing of children rather than adolescents. Indeed, several researchers of longitudinal studies included in this review stopped tracking children’s socio-emotional functioning by the time they reached the upper primary school years, see for example [48,50,55]. This presents a significant gap in the literature. Adolescence is a risk period for the onset of several mental health concerns [145] which may exacerbate for children with dyslexia due to concerns about the changes in the secondary school environment such as the increased difficulty of reading and workload, see for example [39]. To complement the field, future researchers are encouraged to track children with and without dyslexia as they transition from primary to secondary school and beyond. Current research suggests that this school transition may be a risk period for both school connectedness and mental health concerns among children broadly [146,147] and may have specific relevance for children with reading difficulties [46].

### 4.8. Limitations and Future Research

The current review is limited by our exclusion of studies in which reading ability was measured as a continuous variable. We are aware of studies of this kind which would have addressed our research aim, see for example [148]. The decision to exclude these studies was influenced by the feasibility of the review process and the readability of a review with 100+ sources. Additionally, a quality assessment of studies in this field is warranted given the range of methodological limitations which has been briefly discussed. Nevertheless, we identified a variety of individual, family, and community-level factors which may influence mental health in the context of developmental dyslexia, located gaps in the literature, and offered suggestions for future research. Future research which examines associations between aspects of children’s socio-emotional competencies (e.g., emotion regulation), domains of self-esteem, coping strategies, school connectedness, and sub-types of mental health (e.g., depression) in the context of childhood dyslexia is encouraged. This will improve understanding about risk and protective factors for the mental health of children with dyslexia. In terms of methodological factors, our review highlighted a need for more longitudinal work (especially over the transition to secondary school and beyond), which includes dyslexia-specific samples alongside typically developing comparison groups and includes child as well as parent/teacher perspectives wherever possible.

## 5. Conclusions

The current review highlighted a broad range of individual, family, and community-level factors which may influence mental health in the context of developmental dyslexia but relatively few studies which tested associations between third factors and mental health. We identified several gaps in the literature regarding both the content (e.g., school connectedness) and methods (e.g., child perspectives on mental health) of current research and proposed recommendations for future research. Such research is needed to help to identify children who are more (or less) at risk of mental health concerns and to inform tailored mental health promotion programs for children with dyslexia.

## Figures and Tables

**Figure 1 ijerph-20-01653-f001:**
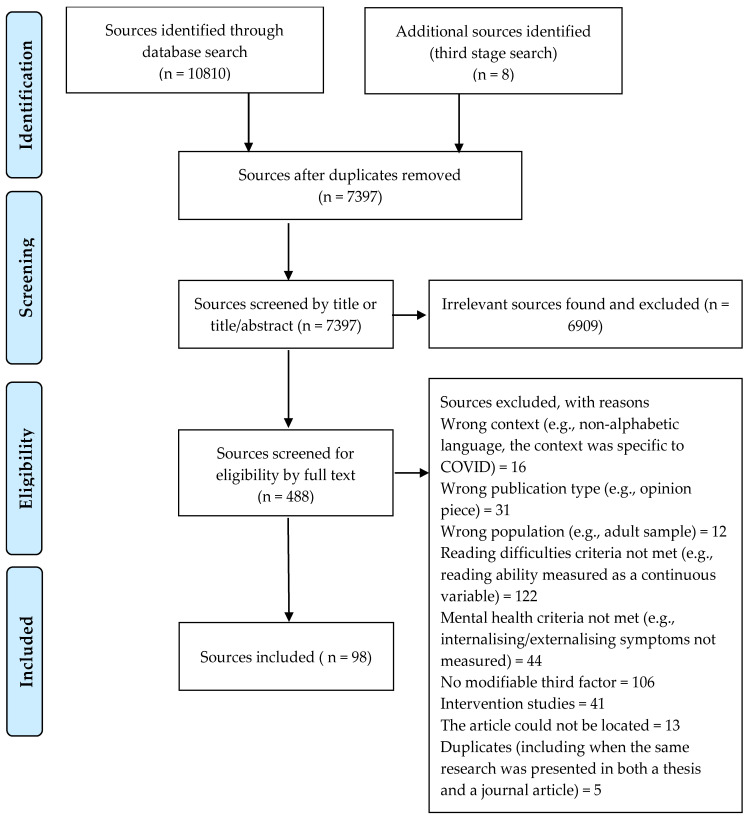
PRISMA flow diagram http://prisma-statement.org/ (accessed on 28 January 2021) and adapted for this scoping review.

**Table 1 ijerph-20-01653-t001:** “Third factors” and associations with internalising/externalising mental health (MH) concerns among children with reading/learning difficulties.

	Third Factor	Qualitative	Quantitative			Total
			Association between Third Factor and MH Found. *	No Association between Third Factor and MH Found.	Third Factor and MH Measured. Association Not Tested.	
Individual-level (70 studies; 71.4%)						
	Self-esteem (including self-perception, self-efficacy, self-worth) ^†^	[14,38,42,68,69,70]	[29,36,39,71,72,73,74,75]		[40,60,65,76,77,78,79,80,81,82,83,84,85,86,87,88]	30
	Stress, coping and resilience (including locus of control and avoidance)	[38,68,69,89,90]	[39,72,91,92]	[93]	[35,81,87,94,95,96,97,98]	18
	Social skills (including language/communication skills and pro-sociality)	[89]	[29,62,89,99]	[74]	[48,50,52,53,54,56,60,63,97,100,101,102,103,104,105]	20
	Emotion regulation (including emotional intelligence and emotional competence)	[69,89]	[29,89,99,106,107]		[97]	7
	Academic performance	[47,77]	[75,85,86,108]		[44,57,62,67,103]	11
	Attitude and approach to learning (including preoccupation with reading difficulties, learned helplessness towards schoolwork, focusing on strengths)	[68,70,90]	[64,109]		[43,52,53,54,57,65,84,103]	13
	Functional impairment				[67,110]	2
	Response to diagnosis and disclosure situations (e.g., shame)	[11,14,38,47,111]				5
	Self-awareness and self-advocacy skills (including understanding dyslexia and taking a strengths-focus)	[89]	[73]			2
	Age of recognition of reading difficulties	[38,70]	[93,112]			4
Family-level (39 studies: 39.8%)						
	Parental mental health and self-esteem	[89]	[83,89,96,99,113,114]	[53,64]	[60,97,115]	11
	Parenting stress, coping, and parenting self-efficacy/confidence		[83,114]	[45,113]	[37,60,66,115]	8
	Parent emotion awareness and regulation			[104]	[97]	2
	Parenting values/practices (e.g., school involvement, “understanding dyslexia”)	[89]	[53,73,83,89,99,116]	[45,95]	[60,63]	10
	Social support for parents (including parent advocacy for support)		[113,116]			2
	Parent-child relationship	[89]	[29,73,89,92,98,99,117,118,119]	[39,45,96]	[43,81]	14
	Parent support for the child	[14,38,42,68,69]	[72]	[39]	[76,88]	9
	Family functioning	[89]	[74]	[89,99,120]	[63,66,103]	7
	Parent preoccupation with child’s reading difficulties		[64]			1
Community-level (54 studies; 55.1%)						
	Teacher-child relationship (e.g., teacher (mis) understanding)	[14,42,90,121]	[29,117,119,122]		[43]	9
	Teacher/school support (e.g., appropriate accommodation/adjustment)	[14,38,123]	[72]		[76,88]	6
	Friendship/peer support	[77,89]	[72,117]		[76,81,88]	7
	Bullying	[14,47,68,69,77,111]	[29,36,58,124,125]			11
	School connectedness (including liking school and school involvement)	[77]	[107,126]	[39]	[65,67]	6
	Stigma/discrimination	[11,14,121]				3
	Social problems (including loneliness)	[68,69]	[29,36,39,73]	[49]	[41,44,46,48,50,51,55,59,60,61,62,67,78,84,96,97,98,102,120,127,128,129]	29

* Check Appendix A for direction of association. ^†^ This category includes many domains of self-esteem (or related constructs), such as social self-esteem, scholastic self-esteem etc. Check Appendix A for specific domain of self-esteem explored in the study.

## Data Availability

The article is a scoping review and no data were collected.

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
