# Peer review of "Understanding Mental Health in Developmental Dyslexia: A Scoping Review"

_ijerph, 2023, doi:10.3390/ijerph20021653_

Round 1

Reviewer 1 Report

Dear Editor and Authors,

I wish to congratulate the authors on an excellent piece of work. I really enjoyed reading this super-clear scoping review. It gives a very nice overview of the research in this area and I believe it is a major contribution that also guides future work. Thank you!

I also believe that this is the most positive reviewer experience for me so far.  I have only two small issues

1.Row 94: I think the date 4.05.2021 should be 04.05.2021

2. Table 1: I would have another major heading above columns 4-6 saying "Quantitative" and in column 4-5 headings instead of "reported" I would say "found" because I was not sure if the columns meant findings or reporting about those.

Author Response

1) Row 94 I think the date 4.08.2021 should be 04.05.2021 (see page 3, row 99).

We thank the reviewer for this observation. In response we have corrected the date as suggested and formatted all dates in the article for consistency (see page 3, rows 9, 142, and 146)

2) Table 1: I would have another major heading above column 4-6 saying “Quantitative” and in column 4-5 headings instead of “reported” I would say “found” because I was not sure if the columns meant findings or reporting about those.

We have made the suggested changes to Table 1 and believe it is much clearer as a result. Additionally, we have added the following lines for clarification regarding the results (see page 7, rows 256-257 and 266-268).

“excluding the adult retrospective studies where adults reflected on their own childhood experiences)” (page 7, row 256-257)

and

"Studies have been placed in a column depending on whether they primarily dealt with the third factor using qualitative, quantitative or both approaches.“ (page 7, row 266-268).

Reviewer 2 Report

A very readable paper, very clear, useful for clinical practices and policies for people with dyslexia and other conditions. Thank you for the meta analysis.

The main question is the association between dyslexia and internalising and externalising mental health concerns. The authors investigated the factors which may influence this association

 Yes, it is very relevant, because the impact of dyslexia is much more than the impact on reading itself. It is about quality of life.

The originality of the paper is based on the analysis of many papers describing mental health topics.

It adds a wider vision of dyslexia, not linguistic, psycholinguistic or other points of view.

The paper is well written and easy and clear to read.

the conclusion are consistent with the evidence and arguments presented. They address the main question.

Author Response

We thank the reviewer for their positive feedback.

Reviewer 3 Report

This is a very interesting and important paper in an under-researched field. The paper is well written and the method part is great work.  I think the paper in its present form is almost ready for publishing, but I would like the authors also to define mental health in the paper.

Author Response

1) I would like the authors to define mental health in the paper.

Thank you for the feedback. As suggested, we have added the following definition to the article (see page 2, rows 72-77).

“For the purpose of this review, mental health concerns are defined according to an internalising/externalising dichotomy (Achenbach, 1966) which is widely used in child development literature and clinical settings. Internalising refers to expressions of emotional distress which are inwardly focused (e.g., withdrawal) whilst externalising refers to expressions of emotional distress which are outwardly focused (e.g., aggression).” (page 2, rows 72-77).